# Transvenous Lead Extraction Procedure—Indications, Methods, and Complications

**DOI:** 10.3390/biomedicines10112780

**Published:** 2022-11-01

**Authors:** Paul-Mihai Boarescu, Adela-Nicoleta Roşian, Ştefan Horia Roşian

**Affiliations:** 1Department of Pharmacology, Toxicology and Clinical Pharmacology, “Iuliu Haţieganu” University of Medicine and Pharmacy Cluj-Napoca, Gheorghe Marinescu Street No. 23, 400337 Cluj-Napoca, Romania; 2“Niculae Stăncioiu” Heart Institute Cluj-Napoca, Calea Moților Street No. 19-21, 400001 Cluj-Napoca, Romania; 3Department of Cardiology—Heart Institute, “Iuliu Haţieganu” University of Medicine and Pharmacy Cluj-Napoca, Calea Moților Street No. 19-21, 400001 Cluj-Napoca, Romania

**Keywords:** transvenous lead extraction, cardiac implantable electronic devices, clinical cases

## Abstract

Transvenous lead extraction (TLE) is a complex and technically challenging procedure useful in the management strategy for many complications related to the presence of cardiac implantable electronic devices (CIEDs). The decision to perform lead extraction should take into consideration the clinical indication for the procedure, risks versus benefits, extractor and team experience, and also patient preference for the procedure. A variety of techniques can be used when performing TLE procedures, such as simple traction, traction devices, and various types of sheaths or snares. TLE is a procedure with a potentially high risk of complications that can be divided into major complications, which require rapid intervention, and minor complications, which are more frequent but are not life-threatening. The present review aims to highlight the indications, contraindications, methods, and complications of transvenous lead extraction procedures.

## 1. Introduction

Transvenous lead extraction (TLE) is a complex procedure frequently used for the treatment of complications related to cardiac implantable electronic devices (CIEDs) [1]. In recent years, the need for CIED removal has steadily increased, because more and more patients with sinus node disease, atrioventricular block, advanced heart failure, and/or at risk of sudden cardiac death received cardiac implantable electronic devices (CIEDs) [2].

At the end of the last century, open-heart surgery, which is a more invasive and expensive method, was initially used to remove leads. Over the past three decades, TLE has evolved and nowadays is regarded as the premier method when lead extraction is indicated. Compared with median sternotomy, used on open-heart surgery, TLE involves an endovascular intervention, which is more amenable for patients with several comorbidities, as is often the case with patients with CIEDs [3].

The decision to perform lead extraction should take into consideration the clinical indication for the procedure and also many other factors such as risks versus benefits, extractor and team experience, and even patient preference [3].

A variety of techniques can be used when performing TLE procedures such as simple traction, traction devices, and various types of sheaths or snares. [4].

TLE procedure is an effective and safe method to remove problematic leads, as major complications that require emergent intervention were reported to be low (up to 2.3% of cases) in even the most experienced hands [5].

The aim of the present review is to highlight the indications, contraindications, methods, and complications of transvenous lead extraction procedures.

## 2. Indications for Lead Extraction

### 2.1. Infection

The strongest indication for TLE is CIED infections [6,7]. CIED infections have become increasingly prevalent, due to the number of CIED implantation procedures and since these procedures are performed in an aging population with multiple comorbidities [8,9]. The multiplication of cardiac pacing centers where staff experience is inadequate can be another explanation [10]. Several scenarios can be encountered in clinical practice.

Isolated pocket infection is regarded as a local infection limited to the generator (‘box’) pocket, not involving the leads of the device. It is clinically characterized by local signs of inflammation, including localized pain, warmth, erythema, unexplained persistent pyrexia, tenderness, wound dehiscence, and discharge from the wound (often purulent), with negative blood cultures [11,12].

Isolated pocket infection must be clinically differentiated from superficial infection of the incision, which involves only the suture of the pocket. It is limited to the skin and subcutaneous tissue and has no communication with the generator pocket. It is also considered a local infection, but it may not require lead extraction [11,12].

Isolated pocket erosion is a chronic skin erosion process resulting from the exposure of the generator and/or lead(s) of CIEDs. It can be associated or not with local signs of infection. Adherence of the skin on the device associated with a thinning and browning of the skin might be suggestive of future skin erosion. If erosion happens, the device should be considered infected, regardless of the mechanism for erosion, as it is usually indicative of infection. Some of the patients with this complication are asymptomatic, while others complain of local pain, and usually blood cultures are negative [13].

Pocket infection with lead/valvular endocarditis is another indication of TLE. For these complications, patients present local signs of pocket infection, associate lead or valvular vegetation(s) and the blood cultures are positive. Additional CIED-related endocarditis criteria should be considered, apart from the modified Duke Criteria presented in The 2015 European Society of Cardiology (ESC) guideline, that are currently used to define endocarditis [14]. One criterion is represented by positive cultures of the extracted lead in the case of patients with prior negative blood cultures. Other criteria are represented by the presence of lead vegetations, and abnormal metabolic activity around the CIED generator and/or leads detected by 18F-fluorodeoxyglucose positron emission tomography/computed tomography or radiolabelled leucocytes single-photon-emission computed tomography/computed tomography [13].

Pocket infection with bacteremia is characteristic of patients who present local signs of pocket infection associated with positive blood cultures but without the presence of any lead or valvular vegetation(s) [13].

CIED-related endocarditis without pocket infection is a situation when patients associate bacteremia without any signs of local signs of pocket infection and who are associated or not with lead or valvular vegetation(s) [13].

Patients who present bacteremia without an obvious source other than the CIED, are considered to associate occult bacteremia with a presumable CIED infection. After the TLE procedure, it is expected that bacteremia will disappear [13].

### 2.2. Lead Dysfunction

Lead dysfunction was reported as being the second most frequent reason for lead extraction in the ELECTRa registry [15]. Lead fracture or insulation failure are the most frequent causes of lead dysfunction, resulting in issues with lead parameters such as impedance, sensing, or capture. In some cases, even if the integrity of the lead is compromised, the electrical parameters may still be normal [16]. In case of lead dysfunction, the options are extracting it or isolating and abandoning the lead in the tissue.

### 2.3. Lead-Related Complications

There are also several complications involving functional leads that may necessitate an extraction: perforation, inadvertent placement of a lead in the arterial system, left atrium, or left ventricle, lead thrombosis or lead fragment embolism, bilateral venous stenosis or occlusion (subclavian; superior vena cava) [17,18].

In approximately 25% of the cases in which ICDs were implanted, various degrees of venous stenosis were reported [19].

Patients who present thromboembolic events attributable to a lead fragment or thrombus on a lead or that cannot be treated by other methods, or those who associate superior vena cava occlusion or stenosis that prevents implantation of a necessary lead, also have indications for lead removal [6,7].

Apart from venoplasty or tunneling, a contra-lateral lead across the chest, another reliable alternative method in patients with venous occlusion, is a lead extraction. It is useful to provide a channel through which new leads can be implanted [20]. Additionally, to avoid lead entrapment, if stenting is required for venous stenosis, its extraction is usually performed [17].

### 2.4. Abandoned Functional Leads

Functional leads may no longer be required, and lead extraction might be necessary in situations such as upgrades from a pacemaker to an ICD, downgrading from dual- to single-chamber systems in cases of patients who developed atrial fibrillation, system relocation for radiotherapy, lack of device indication, or even lead recall with prophylactic revision. In these cases, the options are either abandoning or extracting the leads (to reduce the intravascular lead burden and to avoid future issues).

Patients with abandoned leads can associate asymptomatic venous thrombosis; moreover, the association of multiple leads significantly increases the risk of asymptomatic venous occlusion and also infection [21]. It was observed that, in the case of lead abandonment, there is a risk of electrical interference between active and abandoned leads that can result in the oversensing and inappropriate inhibition of pacing with potentially severe consequences [22].

In patients with an abandoned lead who present electrical interferences with the operation of a CIED system, lead extraction can be useful [6,7].

Some factors that favor extraction rather than abandonment include intravascular crowding (e.g., four or more leads in the target vessel or five or more leads in the SVC), electrical interference between the old lead and new leads, a high likelihood of future intravascular device/lead exchanges (e.g., young age), or a high risk of future lead infection (e.g., immunosuppression, history of endocarditis) [4].

### 2.5. Access to Magnetic Resonance Imaging

Lead removal could be an option for patients to facilitate access to magnetic resonance imaging (MRI) in case they need this examination for the management of associated pathology. For these patients, lead removal is necessary to prevent their abandonment and to allow the implantation of an MRI conditional system [6,7].

The radiofrequency (RF) field generated by the MRI machine is responsible for the main interaction between the MRI and the non-MR-conditional pacemaker. If exposed to an MRI field, the RF field might cause adverse effects such as inappropriate device function, device reset as a result of the interaction with the magnetic reed switch, or it can lead to pacing or sensing problems, changes in sensing or capture thresholds or lead perforation due to heating at the lead tip [23].

For MRI explorations at a field strength of more than 1.5 tesla, the presence of MRI non-conditional leads (functional or dysfunctional, even abandoned) represents a contraindication, especially in pacemaker-dependent patients or when with a high capture threshold or high pacing impedance are present and the scan is at the thoracic level [23]. Therefore, when no other diagnostic alternative tool to MRI is available, extraction of leads may be performed in specific situations and in selected cases. Adequate assessment of the risks and benefits of lead extraction before the procedure is mandatory to allow access to this imaging exam.

### 2.6. Chronic Pain

Chronic pain may be a complaint of patients with CIEDs, and it may be attributed to lead insertion, movement-triggered pain, subclinical infection, or a poorly formed device pocket. It is mandatory to recognize that chronic pain may be a sign of an infection and that extraction can relieve constant and intermittent pain in patients with CIED [6,7,24].

Lead extraction and CIED removal are reasonable for those patients who complain of severe chronic pain when alternative management strategies have failed to resolve the problem [6,7]. For patients with chronic pain, the possibility of deepening the pocket (so-called “plastic surgery of the pocket”) is sometimes considered but it might not eliminate the symptoms and increases the risk of infection in patients with old leads [25].

### 2.7. Other Indications

Prophylactic lead removal may be considered for patients with a certain type of lead that, due to its design or even its failure, represents a potential threat to patients if it remains in place [7]. Lead removal can be necessary even for patients with a CIED location that interferes with radiotherapy for the treatment of malignant tumors [6,7] or in patients who present refractory ventricular arrhythmias after lead placement, or life-threatening arrhythmias secondary to retained leads or lead fragments, most probably as a result of mechanical induced ventricular ectopy [6,26].

## 3. Contraindications for Lead Extraction

In most patients, lead removal is associated with higher benefits than potential risks of the procedure. Relative contraindications to percutaneous TLE include lack of required experienced personnel or equipment, lead placement through a systemic venous atrium or systemic ventricle, concomitant need to remove epicardial lead components, known abnormal placement of leads through other structures than normal cardiac and venous structures (e.g., left ventricle, aorta, or subclavian artery) and large vegetations (e.g., larger than 2.5 cm), where open extraction or preemptive use of a suction catheter should be considered [7].

## 4. Lead Extraction Approach

Lead extraction is usually performed via a superior approach at the lead insertion site, although alternative approaches such as the femoral one, are used in certain situations, or even a combination of methods. As an example, an inferior approach via the femoral vein is necessary when the free lead tip cannot be reached from the implant vein [27]. The percutaneous approach to lead extractions is inherently less invasive and significantly reduces patient morbidity; therefore, it is generally preferred over open surgical extractions. Open surgical approaches are rare and usually reserved for complex and high-risk cases to avoid potentially life-threatening complications encountered during percutaneous extractions. Often such cases require a hybrid approach that combines transvenous lead extraction and open-heart surgery to remove the intracardiac and intravascular portions of the leads [4].

### 4.1. Principles of Transvenous Extraction

#### 4.1.1. Simple Traction

Simple traction of the CIED lead, with non-locking stylets, is the first attempt for transvenous lead extraction. It might be successful in up to 85% of cases when leads remain attached at the tip of the myocardium but move freely within the vein lumen. Infected leads or those with a short lead dwell time, less than two years, seem to be the situations when simple traction proved to be useful. If the dwell time is longer, simple traction is likely to cause different complications or fail because over time fibrosis occurs at multiple levels of the lead. In this case, other methods are required to cut these clamps and remove the leads [28].

#### 4.1.2. Locking Stylets

Locking stylets provide internal support to the cardiac lead targeted for removal, so it plays a very important role in lead extraction procedures. It is necessary to concentrate the force on the resistive hinge [28]. The locking stylet is inserted through the lumen of the lead, is advanced to the tip, and fixes the proximal end to the body of the lead. This allows the application of traction force distally, which is crucial for determining the ease of extraction, whether using either simple traction or specialized sheaths [6].

#### 4.1.3. Mechanical Telescoping Sheaths

These sheaths consist of an inner flexible sheath and an outer rigid sheath constructed in Teflon, polypropylene, or steel. Since they are not powered, during typical use, the application of sufficient tension on the locking stylet, combined with alternating counterclockwise and clockwise motions, allows the advancement of the inner and outer sheath pair. The soft inner sheath is used for advance over the rail of the lead, while the outer sheath, which is more rigid, disrupts the fibrotic tissue that attaches the lead to the vein lumen or the other leads. Appropriate positioning of the lead in the vascular space combined with sufficient traction on the locking stylet allows it to act as a rail and remain within the confines of the vasculature under fluoroscopic guidance [27].

#### 4.1.4. Laser Sheaths

The fiber-optic laser sheath delivers the laser energy to the tip of the sheath. The laser and outer sheaths are passed over the lead body, utilizing the standard techniques of counterpressure and countertraction, until the first binding site is reached [29]. The laser sheaths allow the dissolution of the tissue around the tip of the sheath using photomechanical kinetic energy. This is the result of the molecular bonds’ photochemical destruction and photothermal ablation. Photothermal ablation represents the process of water vaporization and the cells’ rupture. [30]. The ablation of the encapsulating tissue results from the gentle advancement pressure on the laser sheath and from the withdrawal traction on the locking stylet. These two operations combined with excimer laser energy result in and allow the sheaths to advance to the next binding site until the lead tip is released and the lead extracted [27]. When compared with mechanical telescoping sheaths, the laser sheath usage produced shortened extraction times and frequent complete lead removal, without an increase in procedural risk [31].

#### 4.1.5. Electrosurgical Dissection Sheaths

The electrosurgical dissection sheath uses radiofrequency energy to cut through fibrous tissue around leads. This kind of sheath is similar to the cautery tool used in surgery, as it uses two electrodes exposed at the tip of the sheath, allowing linear dissection of adhesions. Compared with the laser sheath, where the dissection is circumferential, the electrosurgical dissection sheath allows only localized dissection, as the application of energy is focalized [32]. The EDS uses radiofrequency energy. Because the electrosurgical dissection sheath seems to be less effective on sutures, calcified areas, or insulation materials, the progress in other mechanical methods has effectively eliminated this kind of sheath from the market [28].

#### 4.1.6. Rotational Mechanical Sheaths

The mechanical dilator sheath set consists of an outer telescoping polymer sheath and an inner plastic sheath with a threaded metal tip. The inner sheath is flexible and has a handle trigger-driven rotational tip at the end, which helps bore through the adhesions. This sheath is mounted on a pistol that mechanically rotates the sheath [33]. The mechanical cutting technology used for this sheet has a bidirectional rotating mechanism that rotates a total of 574 degrees (287 degrees clockwise and 287 degrees counterclockwise). This kind of mechanism, which is capable of cutting through dense and calcific scar tissue, has been proven to be an effective alternative to remove leads from densely scarred venous entry sites and/or severely calcified adhesions [27]. Extraction using rotational mechanical sheaths was reported to be successful in up to 98% of cases with no major complications [34,35].

#### 4.1.7. Femoral Snares

The number of femoral extraction procedures has lately been reduced, as a result of the extraction techniques’ success in using a superior approach [28]. There are two steps in lead removal by the femoral approach. First, a free end of the lead is made available for snaring when the proximal or distal end is pulled into the inferior vena cava. Second, traction is placed on the lead, using the free end for the ultimate removal. At the current time, this is usually the procedure of choice when extraction via the vein of the implant is not achievable, or when it is necessary for the removal of cut leads or lead fragments with free ends free-floating in the pulmonary arteries, heart, or venous system [36]. This TLE extraction method has a high success rate of more than 90% and, perhaps, should be adopted earlier in older or passive-fixation leads that are prone to fracture [28,36]. The femoral approach involves longer procedures and higher fluoroscopy time, but it is as safe and effective as others [37].

#### 4.1.8. Surgical Extraction

Surgical extraction strategically is preferable if a prior extraction procedure has failed, the patient has another reason for cardiac surgery, or when cardiac imaging identifies large lead masses (vegetation or thrombus more than 2.5 cm) [6]. A previous sternotomy or thoracotomy is more challenging and changes the approach of the surgical interventions. For example, in patients with previous coronary artery bypass graft surgery, a lateral thoracotomy instead of a sternotomy may be preferred, as the location of an internal mammary vascular graft is important [38].

A summary of the advantages, disadvantages, and reported rate of success for different TLE methods can be found in Table 1.

The comparative analysis of the literature is hampered by the fact that some authors report the procedural success and other authors report the clinical success. The removal of all targeted leads and lead material from the vascular space in the absence of procedure-related death or any permanently disabling complication is the definition for procedural success [9]. The removal of all targeted leads or retention of a small fragment of the lead (<4 cm) that did not negatively impact the outcome goals of the procedure in the absence of any permanently disabling complication or procedure-related death is the definition for clinical success [9]. For example, for polypropylene mechanical catheters the reported clinical success is 95–97% and the reported procedural success is 93–95% [1,2,9,18,25,41].

Even if, at first impression, simple traction and traction using the locking stylet seem to be the most economical TLE methods, they are not reliable in older leads. Studies focused on evaluating the effectiveness of polypropylene, rotational and laser sheaths involved old leads 8–10 years on average, even 15–20 years and more.

## 5. TLE Organizations and Security Measures

To avoid any major or even life-threatening complications, transvenous lead extraction procedures urge the utmost care and experience, as during the procedure the operator faces the lack of direct visualization along the intravascular route. TLE procedures are not frequently performed in all implantation centers. Only a few high-volume centers can provide opportunities for adequate physician training in this field. A minimum of 40 leads in at least 30 procedures are considered as recommended for training, while a minimum of 15 procedures, with at least 20 leads extracted each year, are necessary for maintaining competency [13].

TLE procedure requires a team-oriented approach that anticipates and plans for all potential scenarios and includes: the physician who performs the extraction, a cardiothoracic surgeon on stand-by, an anesthetist, a person to operate the fluoroscopy or an X-ray technician, scrubbed and non-scrubbed assistants. The location for the procedure (electrophysiology or catheterization or laboratory or the operating room), is less important than the immediate availability of cardiothoracic surgical intervention no later than 10 min after the incident, as the most important factor in preventing death due to a major complication is the time to surgical intervention [27].

An anesthesia cart, vasopressors, other emergency medications and additional emergency equipment such as temporary pacing equipment, transthoracic and/or transesophageal echocardiography, vacuum containers for chest tube drainage, a pericardiocentesis tray, are required to be present at immediate availability in case they are needed [27].

Bilateral peripheral venous access and invasive hemodynamic monitoring obtained from invasive arterial catheters are recommended for the TLE procedure [46].

Femoral venous access is necessary for temporary pacing in pacemaker-dependent patients.

With proper usage and a comprehensive rescue protocol, the endovascular occlusion balloon was proved to have the lifesaving capacity in cases of superior vena cava tears during TLE procedures [47].

## 6. Complications

Major complications associated with TLE primarily arise from severe damage to the myocardium or venous vascular walls. Vascular laceration, haemothorax, thromboembolic events (paradoxical systemic emboli in the presence of an atrial septal defect or patent foramen ovale), cardiac avulsion, pericardial tamponade, and even death are considered major complications. Pericardial tamponade is the most common major complication observed in 2.2% of cases [47]. This major complication can be resolved if treated quickly using a sternotomy and surgical repair, but in rare cases of rapid and massive blood loss, death is often the outcome.

Superior vena cava tears are other lethal complications in transvenous lead extraction. If tears happen above the pericardial reflection, the most frequent result is a large haemothorax (risk of 0.13%), while tears below the pericardial reflection may lead to pericardial effusion (risk of 0.19%) [1].

These injuries may also necessitate urgent sternotomy and surgical repair [4]. Even if it is infrequent, vascular laceration carries a high rate of mortality (50%) and is considered the most catastrophic complication. Tools providing rapid temporary endovascular occlusion of the superior vena cava can be useful before hasty thoracotomy and surgical repair [48].

A recent score (SAFeTY TLE score) was suggested for a better patient selection for lead removal procedures using mechanical tools [49,50]. Authors concluded that this score is useful to predict the potential occurrence of procedure-related major complications, and patients with a SAFeTY TLE score higher than 10 must be considered high-risk patients and should be treated at high-volume centers with surgical backup [50].

Minor complications include bleeding requiring transfusion, pocket hematoma requiring evacuation, pneumothorax necessitating chest tube placement, venous thrombosis requiring medical therapy, pericardial effusion without surgical intervention, and migrated lead fragment without sequelae. Although these events are significant and require rapid intervention, they are usually not life-threatening [47].

Tricuspid valve damage with worsening tricuspid regurgitation is considered another complication. The number of extracted leads, extraction of abandoned lead(s), extraction of leads with redundant loops, and also extraction leads placed in the right ventricle, were reported as the risk factors for tricuspid valve damage and cardiac/vascular wall during TLE [51].

Another complication that is less discussed is the risk of TLE-related infection. The timing of re-intervention is important and seems to correlate with the risk of CIED infections. For example, only generator change is associated with a roughly two-fold risk for infection [52]. By deductions, since TLE procedures have a long-duration time, the risk of infection increases depending on the number of leads extracted and the operative time.

A study performed by Polewczyk et al. revealed that early infections after a TLE procedure are probably due to pocket contamination during CIED implantation-related procedures, while delayed and late systemic infections are most probably correlated with other lead-dependent factors such as intracardiac lead abrasion [53].

## 7. Conclusions

Transvenous lead extraction is a useful procedure in the complex management of patients with cardiac implantable electronic devices if they are associated with complications. Infections are the strongest indication for transvenous lead extraction. Simple traction, traction devices, various types of sheaths or snares are the methods used for transvenous lead extraction procedures. Proper preparation for the procedure and a team-oriented approach in case of major complications due to damage to the venous vascular walls or myocardium, which are life-threatening, are mandatory for rapid intervention.

## Figures and Tables

**Table 1 biomedicines-10-02780-t001:** Advantages, disadvantages, and reported rate of success for different TLE methods.

Method	Advantage	Disadvantage	Reported Rate of Success
Simple traction	– useful for infected leads or leads with a short lead dwell time (less than two years) – no costs – reduced fluoroscopy time	– it is likely to cause different complications if the dwell time is longer than two years – risk of lead-related venous obstruction – loss of venous access for implantation of a new lead	up to 85% [39]
Locking stylets	– provides internal support to the cardiac lead targeted for removal – allows the application of traction force distally – reduced costs – reduced fluoroscopy time	– sharp bends in the lead loops and other irregularities in the lead route limit the ability of the locking stylet to reach the tip of lead – risk of lead-related venous obstruction – loss of venous access for implantation of a new lead	up to 93% [40]
Mechanical telescoping sheaths	– produce a technically demanding but effective technique of mechanical disruption of the fibrosis around the lead – provides venous access for the implantation of a new lead – useful for fibrous adhesions – reduced costs	– high fluoroscopy time – long procedure duration – risk of lead fragmentation and fragment retention – risk of collapse/torsion of sheath due to excessive bending – less useful for dense fibrotic or heavily calcified lesions	98% [41]
Laser sheaths	– instead of tearing the tissue attachments it dissolves them; therefore, this method is useful for fibrous lesions and scar tissue – short procedure duration – provides venous access for the implantation of a new lead – reduced fluoroscopy time	– increased potential risk for perforation because they use increased sheaths stiffness associated with the mechanical outer sheath – ablative effects on heavily calcified lesions are minimal – high costs because of a double investment into the laser sheath itself and in the laser system	94% [42]
Electrosurgical dissection sheaths	– improved precision and diminished risk due to focused and steerable dissection plane – short procedure duration – reduced fluoroscopy time	– less effective on sutures, calcified areas, or insulation materials – high costs	93% [43]
Rotational mechanical sheaths	– the flexible shaft of the sheath reduces the risk of venous perforation – the ability of the mechanical cutting tool to be more effectively in removing calcified and densely fibrotic binding sites – less aggressive tip that reduces the damage to leads, vascular structures, and also myocardial tissue – provides venous access for the implantation of a new lead – reduced fluoroscopy time – moderate costs	– less useful for scar tissue	98% [44]
Femoral snares	– useful for removing lead fragments that may break off during the extraction procedure – moderate costs	– high fluoroscopy time – long-duration procedure – loss of venous access for implantation of a new lead	94% [45]
Surgical extraction	– useful when cardiac imaging identifies large lead masses (vegetation or thrombus more than 2.5 cm) – direct visualization of the lead – complete removal of the lead – provides access for epicardial leads implantation	– very high costs – long-duration procedure – a previous sternotomy or thoracotomy is more challenging regarding the approach	

## Data Availability

Not applicable.

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
