# Peer review of "Transvenous Lead Extraction Procedure—Indications, Methods, and Complications"

_biomedicines, 2022, doi:10.3390/biomedicines10112780_

Round 1

Reviewer 1 Report

The paper concerns the very difficult problem of transvenous lead extraction.  The review is s well written general description of the very difficult cardiological technique, that is transvenous lead extraction.  However, the review is very simplified description of the issue and contains only well known information, that have been published many tomes elsewhere.  Therefore I do not consider the paper to be worthy of publication in the journal with so high IF. Moreover, I do not agree with some information concerning complications related to particular techniques, for example lack of major complications with rotational sheaths or number of complications with laser sheaths (the most important disadvantage of the technique).

Author Response

The paper concerns the very difficult problem of transvenous lead extraction.  The review is s well written general description of the very difficult cardiological technique, that is transvenous lead extraction.  However, the review is very simplified description of the issue and contains only well known information, that have been published many tomes elsewhere.  Therefore I do not consider the paper to be worthy of publication in the journal with so high IF. Moreover, I do not agree with some information concerning complications related to particular techniques, for example lack of major complications with rotational sheaths or number of complications with laser sheaths (the most important disadvantage of the technique).

R: Thank you very much for your kind appreciation! We brought the suggested improvements to the manuscript. A new table was created in order to better present the advantages and disadvantages of each method.

Reviewer 2 Report

The idea of ​​the article is good, but it is highly debatable whether this format is appropriate for a scientific article to be published in a peer-reviewed medical journal. Colleagues, you have created a description that is not bad in its essence, but unfortunately it is rather superficial. Illustrations, tables from literature sources, comparison of different technologies are missing. The reviewer thanks you for your work. Do not sink into pessimism, I recommend that you make significant corrections and corrections and submit the article again.

Author Response

The idea of the article is good, but it is highly debatable whether this format is appropriate for a scientific article to be published in a peer-reviewed medical journal. Colleagues, you have created a description that is not bad in its essence, but unfortunately it is rather superficial. Illustrations, tables from literature sources, comparison of different technologies are missing. The reviewer thanks you for your work. Do not sink into pessimism, I recommend that you make significant corrections and corrections and submit the article again

R: Thank you very much for your kind appreciation! We brought the suggested improvements to the manuscript. A new table was created in order to better present the advantages and disadvantages of each method.

Reviewer 3 Report

A necessary study, conveying the current state of knowledge in the field of lead management, focused mainly on the problem of lead extraction. I have a number of comments, so major revision due to their number.

“TLE is a procedure with a high risk of complications”

Please add “potentially” because some phrases later (Introduction) you conclude that “TLE procedure has been shown to be an effective and safe method”

2.1. Infection

“It is also considered a local infection, but it may not require lead extraction. Isolated pocket infection is clinically characterized by local signs of inflammation, including localized pain, warmth, erythema, unexplained persistent pyrexia, tenderness, wound dehiscence, discharge from the wound (often purulent), with negative blood cultures [9,10]”

Unclearly written. The reader may not notice the difference between the treatment of infection in the suture and infection of the device pocket. Clearly separate these two situations.

2.3. Lead-related complications

When discussing the indications for TLE, I suggest, first of all, using the recommendations of scientific societies (HRS 2009, HRS 2017) and citing of them. Other publications may be cited, but the primary source should be recommendations, which are correctly cited later in this paper.

2.4. Abandoned functional leads

I would just like to add one or two sentences about the potentially negative effects of lead abandonment.

2.6. Chronic pain

Maybe one more sentence for non-professionals that in case of the soreness of the pocket and old leads, the so-called plastic surgery of the pocket itself has no chance to eliminate the symptoms and only increases the risk of infection

2.7. Other indications

Prophylactic lead removal …

Prophylactic lead extraction concerns electrodes that can potentially pose a threat to the patient's health in the future, although at the time of making the decision they are not such (threatener lead (loops, free ending, left heart, lead related TV dysfunction). lead abandonment I include change of pacing mode / upgrading, downgrading, abandoned lead / prevention of abandon abandon (AF, overmuch of leads). Other indications to recapture venous access (symptomatic occlusion, SVC syndrome, lead replacement / upgrading) and other (MRI indication, cancer) etc.

I noticed that the authors intentionally refrained from giving a gradation of indications (class 1, class 1a, class 2a, class 2b and class 3). Using the indication classes allows us to more accurately judge when we may consider lead extraction and when we should simply do so.

1.       Contraindications for lead extraction

Everything is generally correct, it should be noted that this applies to standard TLE conditions. Certain procedures are of course possible to be performed under special organizational conditions by highly experienced personnel.

4.1.1 Simple traction

It should be remembered about the frequent phenomenon of lead-related venous obstruction. Simple traction lead removal may result in loss of venous access for implantation of a new lead or lead for prolonged temporary pacing. Simple traction is a technique for removing infected leads.

4.1.2. Locking stylets

This is all true, but sharp bends in the lead loops and other irregularities in the lead route limit the ability of the locking stylet to reach the tip of lead.

4.1.3. Mechanical telescoping sheaths

I would not emphasize so much the differences in the stiffness of the external and internal sheaths (although they certainly exist), I would point out that the use of a pair of catheters reduces the risk of collapse / torsion of sheath due to excessive bending.

5. Complications 255

Tricuspid valve damage with worsening tricuspid regurgitation is considered as another complication. Number of extracted leads, extraction of abandoned lead (s), extraction of leads with redundant loops, and also extraction leads placed in the right atrium, were reported as the risk factors for tricuspid valve damage and cardiac/vascular wall during TLE [39].

If you write about TV damage only, quote more precisely. Of course, the common risk factors for both types of damage are given correctly, but in the following sentences it is stated that RAA lead extraction has no effect on TV damage and the paragraph focuses on TVD damage.

“The timing of re-intervention is important and seems to 290 correlate with the risk of CIED infections. For example, only generator change is associated with a roughly two-fold risk for infection [40]. By deductions, since TLE procedures have a long duration time, the risk of infection increases depending on the number of leads extracted and the operative time”.

Awkward wording "pro-TLE infection". There are three separate issues. Infections of the new CIED after TLE performed for infection (re-infection), recurrence of infection due to incomplete lead removal, and infection of the CIED after TLE performed for non-infectious reasons. Yes, TLE leads with a long stay in the patient's body and TLE combined with the replacement of left ventricular or His leads take longer and entail an increased risk of infection. If you talk about the timing of infections after TLE, I suggest paying attention to the article: Polewczyk A, Jacheć W, Polewczyk M, Szczęśniak-Stańczyk D, Kutarski A. Early, Delayed and Late Cardiac Implantable Electronic Device Infections: Do the Timing of Onset and Pathogens Matter? J Clin Med. 2022;11:3929

One paragraph is missing: "TLE organizations and security measures", which is to indicate that in the event of serious complications, emergency sternotomy is to be performed at the scene (no patient transport is included) no later than 10 minutes after the incident. The conditions for the earliest possible diagnosis of the complication (monitoring) and its treatment (readiness of the cardiac surgery team) are to be ensured.

6. Conclusions

I suggest reformulating the last conclusion to Because major complications associated with transvenous lead extraction arise from damage to the venous vascular walls or myocardium and they are potentially life threatening the organization of the procedure must give the possibility of mandatory rapid in place of event.

References

The form of representation of references is incorrect (remove DOI and everything after that). The number of authors must be in accordance with the journal's regulations, you may need to limit the number of authors and use the “at al” formula.

Author Response

A necessary study, conveying the current state of knowledge in the field of lead management, focused mainly on the problem of lead extraction. I have a number of comments, so major revision due to their number.

 R: Thank you very much for your kind appreciation!

“TLE is a procedure with a high risk of complications”

Please add “potentially” because some phrases later (Introduction) you conclude that “TLE procedure has been shown to be an effective and safe method”

 R: Thank you very much for your observation! The correction was done according to your observation.

2.1. Infection

“It is also considered a local infection, but it may not require lead extraction. Isolated pocket infection is clinically characterized by local signs of inflammation, including localized pain, warmth, erythema, unexplained persistent pyrexia, tenderness, wound dehiscence, discharge from the wound (often purulent), with negative blood cultures [9,10]”

Unclearly written. The reader may not notice the difference between the treatment of infection in the suture and infection of the device pocket. Clearly separate these two situations.

R: Thank you very much for your suggestion! The paragraph was reorganized so that the readers could better make the difference between isolated pocket infection and superficial infection of the incision.

2.3. Lead-related complications

When discussing the indications for TLE, I suggest, first of all, using the recommendations of scientific societies (HRS 2009, HRS 2017) and citing of them. Other publications may be cited, but the primary source should be recommendations, which are correctly cited later in this paper.

R: Thank you very much for your suggestion! We totally agree with the reviewer. The paragraph was reorganized and the suggested references were primarily used.

2.4. Abandoned functional leads

I would just like to add one or two sentences about the potentially negative effects of lead abandonment.

R: Thank you very much for your suggestion! More information about risks of abandoned leads was added to the manuscript.

2.6. Chronic pain

Maybe one more sentence for non-professionals that in case of the soreness of the pocket and old leads, the so-called plastic surgery of the pocket itself has no chance to eliminate the symptoms and only increases the risk of infection

R: Thank you very much for your suggestion! This idea was added to the manuscript..

2.7. Other indications

Prophylactic lead removal …

Prophylactic lead extraction concerns electrodes that can potentially pose a threat to the patient's health in the future, although at the time of making the decision they are not such (threatener lead (loops, free ending, left heart, lead related TV dysfunction). lead abandonment I include change of pacing mode / upgrading, downgrading, abandoned lead / prevention of abandon abandon (AF, overmuch of leads). Other indications to recapture venous access (symptomatic occlusion, SVC syndrome, lead replacement / upgrading) and other (MRI indication, cancer) etc.

Thank you very much for your suggestion! The suggested information was inserted not only in the “2.7. Other indications” subsection but also into the other subsections from “Indications”

I noticed that the authors intentionally refrained from giving a gradation of indications (class 1, class 1a, class 2a, class 2b and class 3). Using the indication classes allows us to more accurately judge when we may consider lead extraction and when we should simply do so.

 R: Thank you very much for your observation! We totally agree with the reviewer but since this manuscript is just a review not a guideline, our intention was to present the indications not to give recommendations for TLE extractions.

  1. Contraindications for lead extraction

Everything is generally correct, it should be noted that this applies to standard TLE conditions. Certain procedures are of course possible to be performed under special organizational conditions by highly experienced personnel.

R: Thank you for your remark! We mentioned as a contraindication the lack of required experienced personnel or equipment.

4.1.1 Simple traction

It should be remembered about the frequent phenomenon of lead-related venous obstruction. Simple traction lead removal may result in loss of venous access for implantation of a new lead or lead for prolonged temporary pacing. Simple traction is a technique for removing infected leads.

R: Thank you for your suggestion! The new table that we created includes this information.

4.1.2. Locking stylets

This is all true, but sharp bends in the lead loops and other irregularities in the lead route limit the ability of the locking stylet to reach the tip of lead.

R: Thank you for your suggestion! The new table that we created includes this information.

4.1.3. Mechanical telescoping sheaths

I would not emphasize so much the differences in the stiffness of the external and internal sheaths (although they certainly exist), I would point out that the use of a pair of catheters reduces the risk of collapse / torsion of sheath due to excessive bending.

R: Thank you for your suggestion! We are not familiar with the technique described by the reviewer and we did not find information about it. We would be grateful if the reviewer could give us more information or if it is possible to suggest us a reference.

  1. Complications 255

Tricuspid valve damage with worsening tricuspid regurgitation is considered as another complication. Number of extracted leads, extraction of abandoned lead (s), extraction of leads with redundant loops, and also extraction leads placed in the right atrium, were reported as the risk factors for tricuspid valve damage and cardiac/vascular wall during TLE [39].

If you write about TV damage only, quote more precisely. Of course, the common risk factors for both types of damage are given correctly, but in the following sentences it is stated that RAA lead extraction has no effect on TV damage and the paragraph focuses on TVD damage.

 R: Thank you very much for your observation! Indeed the reviewer is right. We apologise for the error. We corrected it.

“The timing of re-intervention is important and seems to 290 correlate with the risk of CIED infections. For example, only generator change is associated with a roughly two-fold risk for infection [40]. By deductions, since TLE procedures have a long duration time, the risk of infection increases depending on the number of leads extracted and the operative time”.

Awkward wording "pro-TLE infection". There are three separate issues. Infections of the new CIED after TLE performed for infection (re-infection), recurrence of infection due to incomplete lead removal, and infection of the CIED after TLE performed for non-infectious reasons. Yes, TLE leads with a long stay in the patient's body and TLE combined with the replacement of left ventricular or His leads take longer and entail an increased risk of infection. If you talk about the timing of infections after TLE, I suggest paying attention to the article: Polewczyk A, Jacheć W, Polewczyk M, Szczęśniak-Stańczyk D, Kutarski A. Early, Delayed and Late Cardiac Implantable Electronic Device Infections: Do the Timing of Onset and Pathogens Matter? J Clin Med. 2022;11:3929

 R: Thank you very much for your observation and suggested reference! We rephrased that part of the manuscript according to the data from the article.

One paragraph is missing: "TLE organizations and security measures", which is to indicate that in the event of serious complications, emergency sternotomy is to be performed at the scene (no patient transport is included) no later than 10 minutes after the incident. The conditions for the earliest possible diagnosis of the complication (monitoring) and its treatment (readiness of the cardiac surgery team) are to be ensured.

R: Thank you for your suggestion! A new paragraph was inserted as suggested.

  1. Conclusions

I suggest reformulating the last conclusion to Because major complications associated with transvenous lead extraction arise from damage to the venous vascular walls or myocardium and they are potentially life threatening the organization of the procedure must give the possibility of mandatory rapid in place of event.

R: Thank you for your suggestion! The conclusion was revised.

References

The form of representation of references is incorrect (remove DOI and everything after that). The number of authors must be in accordance with the journal's regulations, you may need to limit the number of authors and use the “at al” formula.

R: Thank you for your observation. According to the journal’s regulations, it is mentioned that: “Your references may be in any style, provided that you use consistent formatting throughout. It is essential to include author(s) name(s), journal or book title, article or chapter title (where required), year of publication, volume and issue (where appropriate) and pagination. DOI numbers (Digital Object Identifier) are not mandatory but highly encouraged.”

Round 2

Reviewer 1 Report

The new version of the manuscript is much better and more interesting then the older one.  Although the novelty is scarce the paper could be interesting for scientists and cardiologists  not involved in cardiac pacing. The paper required major linguistic and editorial revision.,

Author Response

The new version of the manuscript is much better and more interesting then the older one.  Although the novelty is scarce the paper could be interesting for scientists and cardiologists  not involved in cardiac pacing. The paper required major linguistic and editorial revision.

R: Thank you very much for your kind appreciation! English revision was done by a proficiency English speaker.

Reviewer 2 Report

The authors have made significant changes, corrections and additions to the article. Currently, compared to the previous version, the article is of much higher quality and of higher scientific value. The material has remained much more informative after the changes, and the additions make it more scientifically valuable.

The analyzed topic is extremely relevant and the article will certainly arouse interest among specialists

Author Response

The authors have made significant changes, corrections and additions to the article. Currently, compared to the previous version, the article is of much higher quality and of higher scientific value. The material has remained much more informative after the changes, and the additions make it more scientifically valuable.

The analyzed topic is extremely relevant and the article will certainly arouse interest among specialists

R: Thank you very much for your kind appreciation!

Reviewer 3 Report

Only I have one comment regarding the (valuable) table 1. A non-professional may lead to an erroneous conclusion that the simple traction and traction using the locking stylet are the most economical TLE methods. With simple traction, it was rightly noted that it applies to (isodiametric active fixation) leads up to the age of about 2 years. Anyone who extracts the leads knows that traction using the locking stylet can be effective as well as the simple traction of slightly older, several-year-old leads. And the stated effectiveness applies to such leads. Subsequent tools were evaluated for all leads 6-8-10 years on average, but all studies assessing the effectiveness of polypropylene, rotational and laser sheaths also concerned the oldest leads (15-20 years and more). Hence, only apparent effectiveness seems to be lower.

Additionally, the comparative analysis of the literature is hampered by the fact that some authors report procedural success and some authors clinical success (which allows for the leaving of small fragments of hedgehog leads do not affect the patient's health after TLE). Polypropylene mechanical catheters have efficacy assessed as clinical success 95-97% and procedural success 93-95% [1,2,9,18,25,46,49-53].

Besides, I have no further comments on the manuscript

Author Response

Only I have one comment regarding the (valuable) table 1. A non-professional may lead to an erroneous conclusion that the simple traction and traction using the locking stylet are the most economical TLE methods. With simple traction, it was rightly noted that it applies to (isodiametric active fixation) leads up to the age of about 2 years. Anyone who extracts the leads knows that traction using the locking stylet can be effective as well as the simple traction of slightly older, several-year-old leads. And the stated effectiveness applies to such leads. Subsequent tools were evaluated for all leads 6-8-10 years on average, but all studies assessing the effectiveness of polypropylene, rotational and laser sheaths also concerned the oldest leads (15-20 years and more). Hence, only apparent effectiveness seems to be lower.

R: We are grateful for your observation! We included it in the last version of the manuscript!

Additionally, the comparative analysis of the literature is hampered by the fact that some authors report procedural success and some authors clinical success (which allows for the leaving of small fragments of hedgehog leads do not affect the patient's health after TLE). Polypropylene mechanical catheters have efficacy assessed as clinical success 95-97% and procedural success 93-95% [1,2,9,18,25,46,49-53].

Besides, I have no further comments on the manuscript

R: Tank you very much you your valuable remark !!! We included this idea in the last version of the  manuscript!